# Binaphthyl-Based Chiral Macrocyclic Hosts for the Selective Recognition of Iodide Anions

**DOI:** 10.3390/molecules28124784

**Published:** 2023-06-15

**Authors:** Zong-Cheng Wang, Ying-Zi Tan, Lin-Li Tang, Fei Zeng

**Affiliations:** Department of Biology and Chemistry, Hunan Universityof Science and Engineering, Yongzhou 415199, China; wangzongch@huse.edu.cn (Z.-C.W.); shary8485034@126.com (Y.-Z.T.); lyn870807@126.com (L.-L.T.)

**Keywords:** binaphthyl, chiral macrocyclic, iodide anion, recognition, supramolecular chemistry

## Abstract

In this study, we explorethe synthesis of binaphthyl-based chiral macrocyclic hosts for the first time. They exhibited the selective recognition abilities of iodide anions which can be favored over those of other anions (AcO^−^, NO_3_^−^, ClO_4_^−^, HSO_4_^−^, Br^−^, PF_6_^−^, H_2_PO_4_^−^, BF_4_^−^, and CO_3_F_3_S^−^), as confirmed by UV-vis, HRMS, and ^1^H NMR spectroscopy experiments, as well as DFT calculations. Neutral aryl C–H···anion interactions play an important role in the formation complexes. The recognition process can be observed by the naked eye.

## 1. Introduction

Anions are ubiquitous and play an important role in our body. For example, iodine [1,2,3] is araw material used for the synthesis of thyroid hormones, which can promote the metabolism of substances; regulate the metabolism of proteins, fats, and sugars; and help regulate the metabolism of water and salt. However, iodine deficiency can cause diseases such as goiter or hypothyroidism. However, the most common effects of hyperiodine on thyroid function are iodine-induced goiter (IH) and hyperiodine hyperthyroidism. Moreover, the radioisotopes of ^129^I^−^ and ^130^I^−^ are considered harmful to the environment [4]. Therefore, the development of iodide anion receptors is of great value and has attracted considerable interest.

The continuous synthesis of novel macrocyclic host molecules and their unique molecular recognition properties have driven the development of supramolecular chemistry [5,6,7,8,9,10,11]. During the past several decades, various macrocyclic hosts have been developed and show excellent recognition properties for anions (such as fluoride, nitrate, oxyacid, and other anions). Outstanding examples include Sessler’s calixpyrrole [12,13,14], Farnham’s fluorinated macrocyclic ethers [15], Flood’s triazolophane [16,17,18], Sindelar’sbambusuril macrocycle [19,20,21], Beer’s rotaxanes and catenanes [22,23,24,25], and so on [26,27,28,29,30]. Most strategies involve the modification of cavities by employing hydrogen bonds offered by specific bindingsites to bind anions with size and shape selectivity in various media. However, the purpose of constructing iodide anion receptor macrocyclic [31,32], due the large diameter and low electron density [33], rarely makes it difficult to form hydrogen bonds and iodide anion–π interactions. After neutral C–H···anion interactions were elucidated by chemists in 2008 [34,35,36,37], this kind of interaction has attracted significant interest and has developed rapidly. Flood and co-works successfully prepared triazolophane and cages and found that these triazolophane and cages display strong affinity for anions through neutral C–H···anion interactions [16,17,18]. Li’s group reported anionsin water recognized by cages through neutral C–H···anion interactions [38,39,40]. Huang’s group demonstrated that a preorganized rigid macrocycle cyclo[4]carbazole can act as an iodide anion receptor through neutral C–H···anion interactions [41]. Despite these seminal reports, we still need to explorethe application of neutral C–H···anion interactions, especially for the selective recognition of iodide anions.

Since the aryl C–H groups form stronger hydrogen bonds with anions than alkyl C–H group [42], we explored the possibility of using binaphthyl units containing a large number of aryl C–H groups as building blocks to construct macrocyclic arene that can selectively recognize iodide anionsthrough neutral C–H···anion interactions. Recently, Wang and co-works reported the easy preparation of a series of triazine- and binaphthol-based chiral macrocycles and cages [43]. Inspired by their work, in this study, we prepared enantiopure macrocyclic arene composed of chiral enantiomeric binaphthyl units, named **RR-1** and **SS-1**, respectively (Figure 1). It was found that **RR-1** and **SS-1** have a proper cavity size (approximately 10.100 Å × 9.000 Å). ^1^H NMR and UV-vis experiments demonstrated that **RR-1** and **SS-1** could be used as exclusive selectivity sites for iodide anion receptors relative to other anions (such as AcO^−^, NO_3_^−^, ClO_4_^−^, HSO_4_^−^, Br^−^, PF_6_^−^, H_2_PO_4_^−^, BF_4_^−^, and CO_3_F_3_S^−^). Neutral C–H···anion interactions play an important role in the formation of **RR-1**/iodineand **SS-1**/iodide anion complexes.

## 2. Results and Discussion

The synthesis of **RR-1** and **SS-1** isoutlined in Figure 1. **R-3** and **S-3** were prepared according to the literature [44]. We then obtained **R-2** and **S-2** viathe Suzuki coupling reaction of 2,4-dimethoxybenzeneboronic acid with **R-3** or **S-3** inthe presence of Pd(PPh_3_)_4_ as the catalyst in a 68–70% yield. Finally, **RR-1** and **SS-1** weresynthesized in moderate yield following thetreatment of **R-2** or **S-2** with paraformaldehyde and boron trifluoride diethyl etherate in dichloromethane at room temperature. The structuresof **RR-1** and **SS-1** were confirmed by ^1^H NMR, ^13^C NMR, as well as HRMS spectra. The CD spectra of **RR-1** (black line) and **SS-1** (red line) showed mirror images (Figure 2c), providing strong evidence for the handedness of enantiopure macrocycles.

The attempts to obtain the single crystals of **RR-1** and **SS-1** that are suitable for X-ray analysis ended in failure. Thus, density functional theory (DFT) methods were used to gain further insightinto the structures of **RR-1** and **SS-1** by usingGaussian 09 software and by choosing 6-311G as the basis sets. As shown in Figure 2, both **RR-1** and **SS-1** hadbox-like structures with a cavity size of approximately 10.100 Å × 9.000 Å. Initially, we hypothesized that **RR-1** and **SS-1** contain electron-rich cavities that could be used for complexes with cationic guest molecules. Unfortunately, when **RR-1** (4.0 mM) and 1.0 equiv. tetramethylammonium hexafluorophosphate were mixed in CDCl_3_/CD_3_CN (*v*/*v* = 1/1), the proton signals of both **RR-1** and tetramethylammonium hexafluorophosphate were not shifted, suggesting that no complexation occurred between **RR-1** and tetramethylammonium hexafluorophosphate (Appendix A). Since **RR-1** and **SS-1** contain a large number of neutral aryl C–H bonds, we questioned whether **RR-1** and **SS-1** can be used as receptorsfor anions.

Consequently, UV-vis experiments were carried out to verify our hypothesis and commercially available tetrabutylammonium salts (TBAX) were used as anion sourcesdue totheir simple composition and good solubility in chloroform. As shown in Figure 3a, after the addition of 5.0 equiv. tetrabutylammonium salts (TBAX, X = AcO^−^, NO_3_^−^, ClO_4_^−^, HSO_4_^−^, Br^−^, I^−^, PF_6_^−^, H_2_PO_4_^−^, BF_4_^−^, and CO_3_F_3_S^−^) to the solution of **RR-1** in chloroform, the color of the solution containing **RR-1** and TBAI changed from colorless to light orange, whereas the others remained colorless. This obvious color change suggests that interactions between **RR-1** and TBAI may have occurred. UV-vis experiments further reveal the interactions behavior between **RR-1** and TBAX. Upon the addition of 5.0 equiv TBAI, the absorption at 300 nm and 350 nm was enhanced significantly, and new absorption bands appeared at 375 nm, indicating the formation of **RR-1/**iodide complexes in the solution. On the other hand, no absorption spectral changes were observed after the addition of other TBAX salts mentioned above. All the above results indicate that **RR-1** has the ability toselectively recognize iodide anionsover other tested anions. Similar to **RR-1**, **SS-1** also showedthe selective recognition of iodide anionsover other tested anions (Figure 3b).

^1^H NMR experiments further provided the evidence for the formation of **RR-1/**iodide complexes in the solution. After **RR-1** and TBAI with a 1:1 molar ratio were mixed in CDCl_3_, a new set of proton signals that differedfrom **RR-1** and TBAI were observed on the^1^H NMR spectrum, indicating that a new complex **RR-1/**iodide was formed (Figure 4). The protons b and e corresponding to the **RR-1** shift up-field by 0.006 and 0.004 ppm, respectively, which could be attributed to the formation of neutral C–H···anion interactions between **RR-1** and TBAI. Only the protons b and e corresponding to **RR-1** wereshifted, leading us to doubt that the recognition of iodide using **RR-1** may occur on the outside of the cavity. Moreover, unlike the recognition of iodide usingcyclo[4]carbazole [41], which is a slow process and occurs inside the cavity, the complex and decomplex between **RR-1** and TBAI is a fast exchange process on the NMR time scale at room temperature. This differencemay be due to the iodide being recognized by **RR-1** through neutral C–H···anion interactions that are outside of the cavity. To further obtaininsights into the complexation process between **RR-1** and TBAI, ^1^H NMR spectroscopic titrations experiments were then carried out. By monitoring the change inproton b corresponding to **RR-1** followingthe addition of TBAI, a1:1 complex between **RR-1** and TBAI was formed by the mole ratio plot. The binding constant *K_a_* of the complex **RR-1/**iodide was determined to be 132.8 ± 33.8 M^−1^ using BindFit software (http://supramolecular.org). Consequently, the 1:1 complex of **SS-1/**iodide was also formed and the binding constant was calculated to be *K_a_* = 119.1 ± 32.6 M^−1^ (Appendix A).

The energy-minimized optimized structure of **RR-1/**iodide and **SS-1/**iodide further supported the formation of neutral C–H···anion interactions. As shown in Figure 5a, the iodide anion waslocated outside the cavity of **RR-1** through C–H···anion interactions with distances of 3.307 and 3.123 Å, respectively. Only protons b and e corresponding to **RR-1** participated in the formation of hydrogen bonds with iodide anions, which is consistent with the results in ^1^H NMR experiments indicating that only the protons b and e shifted up-field. In the structure of the complex **SS-1/**iodide, an iodide anionwasalso located outside the cavity of **SS-1** through C–H···anion interactions with distances of 3.307 and 3.123 Å, respectively (Figure 5b).

## 3. Materials and Methods

### 3.1. General Considerations

Unless otherwise noted, all reagents were obtained from commercial suppliers and used without further purification. ^1^H NMR and^13^C NMR spectra were recorded witha Bruker DMX400 NMR spectrometer. Electrospray ionization mass spectra (ESI-MS) were recorded on the Thermo Fisher^®^ Exactive LC-MS spectrometer (Thermo Fisher Scientific, Waltham, MA, USA).

### 3.2. Typical Procedure for the Synthesis of RR-1

To a mixture of **R-2** (1.17 g, 2.0 mmol) and paraformaldehyde (180 mg, 6.0 mmol) in dichloromethane (150 mL),boron trifluoride diethyl etherate (0.3 mL, 2.4 mmol) was added. The mixture was stirred at room temperature for 0.5 h. Then, the reaction was quenched by the addition of 150 mL of water. The organic layer was separated and dried with anhydrous MgSO_4_. The solvent was removed in vacuo and the residue was separated viacolumn chromatography on silica gel (eluent: 2:1 DCM/petroleum ether) to give **RR-1** (538 mg, 45%) as a yellow solid. ^1^H NMR (400 MHz, Chloroform-*d*) δ 7.90 (d, *J* = 9.0 Hz, 4H), 7.84 (s, 4H), 7.39 (d, *J* = 9.0 Hz, 4H), 7.26 (s, 4H), 7.01–6.93 (m, 8H), 6.60 (s, 4H), 3.97 (s, 4H), 3.93 (s, 12H), 3.81 (s, 12H), 3.76 (s, 12H). ^13^C NMR (101 MHz, CDCl_3_) δ 157.6, 155.8, 154.8, 133.7, 132.7, 132.1, 129.3, 129.2, 128.7, 127.5, 124.6, 122.7, 121.4, 119.6, 114.1, 96.0, 57.0, 56.0, 55.9, 27.8. HRMS (APCI) *m*/*z*: [M+Na]^+^ calculatedfor C_78_H_68_O_12_Na, 1219.4608; found, 1219.4559.

## 4. Conclusions

In summary, we successfully designed and synthesized enantiopure macrocyclic arene **RR-1** and **SS-1** composed of chiral enantiomeric binaphthyl units. The anion receptor ability of **RR-1** and **SS-1** was investigated using UV-vis and ^1^H NMR experiments. **RR-1** and **SS-1** were found to be able to selectivity bind the iodide anion in a 1:1 manner among the ten tested anions, and neutral C–H···anion interactions were found to play an important role in the formation of the **RR-1/**iodideand **SS-1**/iodide anion complex. The complexation between the iodide with **RR-1** or **SS-1** can be observed by the naked eye, and the color of solution changes from colorless to light orange. We believe that this new kind of iodide receptor will pavethe way to the design of new anion receptors via neutral C–H···anion interactions.

## Data Availability

Not applicable.

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
