# Peer review of "Binaphthyl-Based Chiral Macrocyclic Hosts for the Selective Recognition of Iodide Anions"

_molecules, 2023, doi:10.3390/molecules28124784_

Round 1

Reviewer 1 Report

The manuscript describes the synthesis of new compounds which reveal the interesting anion binding mode consisting in the CH...Anion interaction. What is of importance, the binding mode has some kind of selectivity, which deserve to be studied separately in future.

The synthesis procedure is well described, the experiments on the interaction with TBA slats have been carried out carefully.

Data treatment has been done according to the accepted methods.

There is no item to point out as the shortcoming.

 I expect that paper deserve to be published as it is.

Author Response

There are no questions to answer.

Thank you and the reviewers for the valuable comments and suggestions again.

Sincerely yours,

Fei Zeng

Hunan University of Science and Engineering

Yongzhou, 425199

China

Tel: 86-15869977707

Email: zengfei@iccas.ac.cn

Reviewer 2 Report

Comments:

The original paper by Zeng and co-workers developed binaphthyl-based chiral macrocyclic hosts for selective recognition of iodide anion. This is carefully done study and the findings are of great considerable interest. However, a number of points need clarifying and certain statements require further justification.

Questions:

1.     Chiral macrocyclic hosts based on binaphthyl with different substituents are suggested to discuss in the manuscript.

2.     The author mentioned that “Upon addition of 5.0 equiv TBAI, the absorption at 300 nm and 350 nm was enhanced significantly, and new absorption bands appeared at 375 nm, indicating the formation of RR-1/iodide complexes in the solution.” Please explain the changes of those peaks.

3.     There are two “of” in line 72. Please delete one of them.

4.     The title of Figure 2 “a) and b) The energy-minimized structures of RR-1 and SS-1 simulated by Gaussian computer program” is suggested to change into “The energy-minimized structures of RR-1 a) and SS-1 b) simulated by Gaussian computer program

Based on the content and quality of this paper, I think this work is suitable for Molecules. However, the above problems should be addressed before accepting this manuscript for publication.

 Minor editing of English language required

Author Response

1.  According to the suggestion, the discussion of chiral macrocyclic hosts based on binaphthyl has been added in the manuscript.

  1. The absorption band at 300 nm and 350 nm ascended accompanied by a new absorption band at 375 nm, manifesting that a new moiety formed in solution, which may be due to the formation of neutral C–H…anion interactions between RR-1 and TBAI.

3.  According to the suggestion, one “of” has been deleted.4.  According to the suggestion, “a) and b) The energy-minimized structures of RR-1 and SS-1 simulated by Gaussian computer program” has been changed to “The energy-minimized structures of RR-1 a) and SS-1 b) simulated by Gaussian computer program”

Thank you and the reviewers for the valuable comments and suggestions again.

Sincerely yours,

Fei Zeng

Hunan University of Science and Engineering

Yongzhou, 425199

China

Tel: 86-15869977707

Email: zengfei@iccas.ac.cn

Reviewer 3 Report

   The manuscript by Zeng et. al have reported the synthesis of binaphthyl-based chiral macrocyclic hosts and investigated their as anion acceptors for selectively bind the iodide anion among other anions including AcO-, NO3-, ClO4-, HSO4-, Br-, PF6-, H2PO4-, BF4-, CO3F3S-), neutral C–Hanion interactions play an important role in the formation of complex. The results reported by this manuscript are significance in the supramolecular chemistry and would be in interested for the readership in Molecules. Thus, I recommend this paper to be accepted after attention to a few minor points noted below:

1.The figure index of corresponding NMR spectrum analysis of complex SS-1/iodide should be added in the main text.

2.Why a chiral molecule is used to sense an anion?

Minor editing of English language required

Author Response

1.  According to the suggestion, the figure index of corresponding NMR spectrum analysis of complex SS-1/iodide has been added in the manuscript (Figure S9).

  1. Initially, we hypothesized that RR-1 and SS-1 contain electron-rich cavities that could be used for complexes with chiral cationic guest molecules through cation…π interactions, but failed. Considering that the RR-1 and SS-1 contain a large number of neutral aryl C−H bonds, we wonder whether RR-1 and SS-1 can be used as receptor for anions through neutral C–Hanion interactions. Then, we did this work.

Thank you and the reviewers for the valuable comments and suggestions again.

Sincerely yours,

Fei Zeng

Hunan University of Science and Engineering

Yongzhou, 425199

China

Tel: 86-15869977707

Email: zengfei@iccas.ac.cn

Reviewer 4 Report

In the reviewed paper, the authors described the synthesis of chiral macrocycles. The chemical structure of the compounds was confirmed by 1H and 13C NMR and HRMS spectrometry. For new compounds, elemental analysis is also typically required to confirm purity or at least high performance chromatography. The characteristics of the final products should be completed. I'm not entirely sure about the structure of the complex. Small changes in chemical shifts are not necessarily due to the described interactions. The description shows that the complex is not neutral and has a net negative charge. What evidence do the authors have that this is indeed the case? Does the aliphatic part of the spectrum of the complex contain no signals from TBA, if this abbreviation stands for tetrabutylammonium cation? Maybe the ion pair undergoes complexation? In my opinion, ROESY and DOSY experiments should be additionally performed to confirm the existence of the complex.

Round 2

Reviewer 2 Report

I suggested this manuscript can be published in this journal.

Minor editing of English language required.

Reviewer 4 Report

Based on the supplemented data, I have no further comments and accept the publication in its current form.